# Vulnerability of Satellite Quantum Key Distribution to Disruption from Ground-Based Lasers

**DOI:** 10.3390/s21237904

**Published:** 2021-11-26

**Authors:** David R. Gozzard, Shane Walsh, Till Weinhold

**Affiliations:** 1International Space Centre, The University of Western Australia, Perth 6009, Australia; shane.walsh@uwa.edu.au; 2Australian Research Council Centre of Excellence for Engineered Quantum Systems, Department of Physics, The University of Western Australia, Perth 6009, Australia; 3Australian Research Council Centre of Excellence for Engineered Quantum Systems, School of Mathematics and Physics, The University of Queensland, Brisbane 4072, Australia; t.weinhold@uq.edu.au

**Keywords:** quantum key distribution, satellite, security

## Abstract

Satellite-mediated quantum key distribution (QKD) is set to become a critical technology for quantum-secure communication over long distances. While satellite QKD cannot be effectively eavesdropped, we show it can be disrupted (or ‘jammed’) with relatively simple and readily available equipment. We developed an atmospheric attenuation and satellite optical scattering model to estimate the rate of excess noise photons that can be injected into a satellite QKD channel by an off-axis laser, and calculated the effect this added noise has on the quantum bit error rate. We show that a ground-based laser on the order of 1 kW can significantly disrupt modern satellite QKD systems due to photons scattering off the satellite being detected by the QKD receiver on the ground. This class of laser can be purchased commercially, meaning such a method of disruption could be a serious threat to effectively securing high-value communications via satellite QKD in the future. We also discuss these results in relation to likely future developments in satellite-mediated QKD systems, and countermeasures that can be taken against this, and related methods, of disruption.

## 1. Introduction

Rapid developments in quantum computing threaten the forward security of public key cryptography that currently underpins secure communication and trade. Quantum key distribution (QKD) [1] uses the laws of quantum mechanics to provide a guaranteed information-secure cryptographic key exchange between two parties. As QKD technology matures to enable practical key rates over long distances, it will become of extreme importance to areas such as finance, government operations, and defence.

A great deal of effort is being expended in developing practical QKD technologies around the world, and rapid progress is being made on many fronts [2]. However, due to link attenuation and the impossibility of noiselessly amplifying quantum states, QKD via terrestrial optical fibre links is currently limited to a few hundred kilometres [3,4] and extending this limit is extremely challenging [5]. Quantum repeaters, that extend the reach of QKD systems via entanglement swapping [6], are being developed, but these technologies have not yet matured to a practical level.

Satellite-mediated QKD has been demonstrated as the most promising method for creating practical long-distance (on the order of thousands of kilometres) QKD links [7,8]. Optical losses on the free-space link are dominated by geometric factors due to the spreading of the optical beam, while fibre-based QKD systems are limited by absorption, scattering, and quantum state decoherence in the glass fibre. The *Micius* satellite [7], launched in 2016, achieved an average quantum key rate of 1.1 kb/s over a maximum distance of 1200 km, with a quantum channel efficiency 20 orders of magnitude greater than for an optical fibre link of the same maximum length. Advances in other areas of QKD research and development, such as practical quantum repeaters and quantum memories, will be directly applicable to satellite-mediated QKD links, meaning satellite QKD is likely to remain at the forefront of practical QKD link deployment for many years.

While the quantum security of a QKD link between a satellite and ground station is beyond doubt [8], the high-value nature of communications secured via QKD means that attempts to disrupt (or ‘jam’) the transmission of the quantum key by malicious parties are a possibility. In this paper, we show that ground-based lasers of moderate power (on the order of 1 kW depending on the link architecture and geometry) reflecting off the satellite add sufficient excess optical noise to completely disrupt QKD transmissions from low Earth orbit satellites. Today, laser amplifiers on the order of a few kW are available commercial off-the-shelf. Along with the necessary optics and tracking mount, a system able to target a satellite and disrupt its QKD transmission can be assembled for under $200,000 USD, meaning that the means to disrupt satellite QKD are available to even small groups with modest resources. This poses a serious threat to the use of satellite-mediated QKD in the future, and we identify measures that future satellite QKD systems could employ to reduce their susceptibility to this type of disruption.

## 2. Methods

The operation of a QKD link is highly susceptible to attenuation of the transmitted signal, and to the addition of due to because of the extreme sensitivity of the QKD receiver. Because of this sensitivity, the QKD link from *Micius* is unable to operate during the day due to background light, and *Micius* experienced a reduced key rate (increase in quantum bit error rate, QBER, from approximately 0.01 to 0.03) as a result of light pollution from Beijing 110 km away [7]. We show that the extreme sensitivity of the QKD receiver to background noise (excess photons) means that photons from a ground-based laser, scattered from the transmitting satellite, can be detected by the QKD receiver on the ground and reduce, or completely disrupt, the generation of the quantum key.

Because of the large parameter space for future satellite QKD architectures (e.g., orbital altitudes, QKD protocol and future technology, and satellite geometry), we use the *Micius* satellite as a case study to determine what effect optical power levels of 100 W and 1 kW launched from ground-based laser terminals have on reducing, or completely disrupting, the QKD.

We semi-empirically develop a scattering model for the QKD satellite. We then use the orbital dynamics of the satellite and atmospheric free-space link loss calculations to estimate the fraction of photons launched from a ground based laser, reflected off the satellite, that are received at the QKD ground station. We then analyse what effect this excess photon rate has on reducing the generation of a secure quantum key.

### 2.1. Satellite Architecture

Because of the large range of possible QKD satellite designs, for the purposes of this analysis we assume that the hypothetical satellite targeted by the ground-based laser is very similar to *Micius*, i.e., it has the same QKD protocol and implementation, similar satellite bus design, and is in a circular orbit at an altitude of 500 km. Table 1 lists key aspects of the design of *Micius* that are used to inform our analysis.

### 2.2. Satellite Optical Scattering Model

While the QKD link is active, the QKD ground station (henceforth referred to as ‘the ground station’) will have its receiving optic trained at the satellite, and the satellite will have its transmitting optic trained at the ground station. The disrupting laser beam launched from the belligerent laser terminal (henceforth referred to as ‘the laser terminal’) will be highly off-axis compared with the path of the QKD link because the laser terminal is likely to be many kilometres away from the ground station. Because of this, it is necessary to develop a scattering model for the satellite in order to estimate the fraction of photons from the laser terminal reflected in the direction of the ground station.

As noted in Table 1, *Micius* is coated in metallized polymer blankets typically used for thermal shielding of satellites. This material is not smooth, creating a large number of facets to scatter the incident laser light. Additionally, the geometry of the satellite bus and protrusions such as solar panels and antennae will create additional scattering surfaces.

We performed scattering measurements from a wrinkled gold-foil target in the laboratory in order to determine the approximate scattering profile of the satellite. A simplifed schematic of the experimental setup is shown in Figure 1. Laser light was launched into free-space using a 50 mm diameter telescope with an output beam diameter of 34 mm. The laser light reflected off the gold-foil target (simulating the satellite) and into a second 50 mm telescope attached to a single photon detector (SPD) via multi-mode optical fibre. The SPD measured the rate of photons scattered off the target and coupled into the telescope. The first telescope (launch optic) simulated the laser terminal and was swept through an angle of attack, relative to the line between the target and the second telescope (receiving optic), of 5° to 90°. The receiving optic simulated the QKD ground station telescope. The launch optic was kept consistently 1.00 m from the target, while the receiving optic was 2.12 m from the target. These distances were the largest possible, limited by available space in the laboratory. White card stock was also used as a target to provide a near-Lambertian reflector against which to compare the scattering profile of the gold foil. The flat face of the target was oriented towards the receiving optic, simulating the coarse bodily pointing of QKD satellites towards their ground stations.

Future QKD satellites will likely incorporate quantum memories or quantum repeaters, meaning that the satellite will include a receiving optic. Photons from the laser terminal entering the satellite’s receiving optic off-axis can scatter inside the telescope and couple into the QKD receiver on board the satellite. We also measured the internal scattering of our 50 mm telescopes to assess this possible disruption scenario. The launch optic was pointed at the receiving optic and swept through of 0° to 90° relative to the pointing direction of the receiving optic at a consistent distance of 1.12 m. Our 50 mm telescopes have a clear aperture of 48 mm and minimal internal optical baffles.

The detected photon rates in both sets of tests were recorded using the SPD, simulating the QKD receiver, and used to build approximate scattering cross sections for each scenario in order to inform the optical loss estimates.

### 2.3. Orbital Dynamics

The relative location of the ground station and laser terminal, and the angles formed between the ground station, satellite, and laser terminal will determine both the optical loss due to diffraction and atmospheric attenuation, as well as the period during which the satellite is in view of both the ground station and laser terminal, and so the fraction of the QKD transmission period for which the laser terminal is able to disrupt the QKD.

Detailed simulations of the orbit of the satellite, using the orbital track of *Micius* as an example, were performed to determine the effective range of the laser terminal, and the optical losses from laser terminal, to satellite, to ground station.

### 2.4. Free-Space Link Loss

The rate of excess noise photons, $n_{s}$, detected by the QKD ground station after reflecting off the QKD satellite can be calculated from the following equation adapted from [10]
(1)$$n_{s}=\frac{P_{0}\lambda }{hc}\frac{2}{\pi {\left(\theta_{d}R_{1}\right)}^{2}}\exp\left[-2{\left(\frac{\theta_{p}}{\theta_{d}}\right)}^{2}\right]\left[\frac{1}{1+{\left(\theta_{j}/\theta_{d}\right)}^{2}}\right]\left(\frac{\sigma_{sat}A_{r}}{4\pi R_{2}^{2}}\right)\eta_{r}T_{a}T_{c},$$
where $P_{0}$ is the launch power of the laser terminal, λ is the wavelength of the laser, hc is Planck’s constant multiplied by the speed of light, $\theta_{d}$ is the divergence of the laser beam, $R_{1}$ is the distance from the laser terminal to the satellite, $\theta_{p}$ is the pointing error of the laser terminal, $\theta_{j}$ is the RMS jitter of the terminal pointing, $\sigma_{sat}$ is the optical scattering cross section of the satellite, $A_{r}$ is the area of the QKD ground station receiving telescope, $R_{2}$ is the distance from the satellite to the ground station, $\eta_{r}$ is the optical throughout efficiency of the ground station, $T_{a}$ is the transmissivity of the atmosphere at the laser frequency, and $T_{c}$ is the transmissivity of high cirrus cloud.

In general, Equation (Equation 1) takes form: received photon rate equals the launched photon rate, times the launch optic gain, times the pointing error losses, times the receiving optic gain, times the propagation path losses.

For the case where the mobile laser terminal is directly targeting a QKD receiver on board a satellite, Equation (Equation 1) becomes
(2)$$n_{s}=\frac{P_{0}\lambda }{hc}\frac{2}{\pi {\left(\theta_{d}R_{1}\right)}^{2}}\exp\left[-2{\left(\frac{\theta_{p}}{\theta_{d}}\right)}^{2}\right]\left[\frac{1}{1+{\left(\theta_{j}/\theta_{d}\right)}^{2}}\right]\sigma_{rec}T_{a}T_{c},$$
where $\sigma_{rec}$ represents the cross section for scattering of laser photons into the receiving optic on board the satellite.

All of the parameters in Equations (Equation 1) and (Equation 2) are well characterized thanks to conventional satellite laser ranging operations, except for $\sigma_{sat}$ and $\sigma_{rec}$. The experimental determination of the optical scattering profiles is outlined in the previous section. In this analysis, we will assume the laser terminal launches a single-mode Gaussian beam. This is a reasonable assumption because modern commercial off-the-shelf lasers are able to launch on the order of several kilowatts of power with beam quality factors near unity.

The divergence of the laser beam, $\theta_{d}$, and pointing jitter, $\theta_{j}$, are heavily influenced by turbulence in the lower atmosphere. For the purposes of this analysis, we assume that the laser terminal has very good tracking and the loss due to pointing error, $\theta_{p}$, is much smaller than those due to divergence and pointing jitter, so we set $\theta_{p}=0$. The divergence is given by,
(3)$$\theta_{d}=\sqrt{{\left(\frac{4\lambda }{\pi D_{L}}\right)}^{2}+{\left(\frac{2\lambda }{\pi r_{0}}\right)}^{2}},$$
where $D_{L}$ is the diameter of the laser terminal aperture, and $r_{0}$ is the Fried parameter (coherence scale) of the atmosphere, while the turbulence-induced pointing jitter is given by,
(4)$$\theta_{j}=\frac{2.92}{D_{L}^{1/3}}\left(\frac{1}{\sin\theta_{a}}\right)\int_{h}C_{n}^{2}\left(h\right)\;dh,$$
where $\theta_{a}$ is the elevation angle of the satellite from the laser terminal, and $C_{n}^{2}\left(h\right)$ is the turbulence profile of the atmosphere with height *h*.

The turbulence profile is given by the commonly-used Hufnagel-Valley profile [11]:(5)$$C_{n}^{2}\left(h\right)=0.00594{\left(\frac{w}{27}\right)}^{2}{\left(h\times 10^{-5}\right)}^{10}\exp\left(\frac{-h}{1000}\right)+2.7\times 10^{-16}\exp\left(\frac{-5}{1500}\right)+A\exp\left(\frac{-h}{100}\right),$$
where *h* is the height above the ground (in metres), *A* is the value of $C_{n}^{2}$ at ground level, and *w* is the ‘pseudowind’ given by
(6)$$w=\sqrt{\frac{1}{15\times 10^{3}}\int_{5\times 10^{3}}^{20\times 10^{3}}V^{2}\left(h\right)\;dh},$$
where V(h) is given by the Bufton wind model as
(7)$$V\left(h\right)=w_{s}h+V_{g}+30\exp\left[-{\left(\frac{h-9400}{4800}\right)}^{2}\right],$$
where $w_{s}$ is the slew rate of the ground station and $V_{g}$ is the ground wind speed.

This optical loss model is used in conjunction with the scattering results and orbital dynmics calculations to estimate the photon rate reaching the QKD receiver.

### 2.5. Impact on Quantum Bit Error Rate

QBER is a measure that identifies the fraction of signals received during the QKD transmission that were not from the originally transmitted QKD protocol. Such events have the potential to be the result of an eavesdropper performing, for example, a measure and resend attack. As such, once the QBER reaches above a threshold level for the particular QKD protocol, the amount of leaked information to a potential eavesdropper reaches a level where, even with privacy amplification and distillation steps, no provably secure key can be generated. For the BB84 protocol used by *Micius*, this threshold is at 0.11, or 11% [12] and is further reduced if imperfect sources and detection systems [13,14] are considered as all leaked information has to be considered. For our investigation we will focus on the fundamental limitation of an 11% QBER which is independent of any specifics of the utilised equipment or operating conditions.

The QBER, *Q*, is calculated as in [15]:(8)$$Q=\frac{N_{I}}{N_{I}+N_{C}},$$
where $N_{I}$ are all detected events not from the QKD transmission, and $N_{C}$ are the detected events from the protocol. The net key rate can then be calculated from
(9)$$R_{net}=\left[1-f_{p}H_{2}\left(Q\right)\right]R_{sifted},$$
where $f_{p}$ is the inefficiency function for the error correction relative to the theoretical Shannon Limit and $H_{2}\left(Q\right)$ is the binary entropy function
(10)$$H_{2}\left(Q\right)=-Q\log_{2}\left(Q\right)-\left(1-Q\right)\log_{2}\left(1-Q\right).$$

It can be seen that, therefore, the increase in noise photons at low rates leads to a rapid increase in QBER, in turn reducing the available key rate and, once the ratio between signal to noise detections exceeds 8.06, the QBER will exceed the threshold to suppress all BB84 secure key generation. Mailloux [16] provides a more comprehensive analysis of the impact of the equipment on weak state coherent decoy state QKD as used by [7].

## 3. Results

### 3.1. Satellite Optical Scattering Profile

The profile of the optical power scattered from the wrinkled gold foil was highly variable due to the launched beam finding more or less favourable facets on the target. Repeated tests with different scales of wrinkles and translating the target found the profile to average to an approximately $\sqrt{\cos\theta_{a}}$ relation, where $\theta_{a}$ is the angle of incidence of the laser relative to the line between the target and receiving optic. This indicates broader scattering than predicted by assuming a Lambertian reflectance and suggests that the effectiveness of this method of disruption will be more effective at high angles of attack (larger distances between the laser terminal and the ground station) than first expected. However, this also means that less light than expected will reflect towards the ground station at low angles of incidence.

With this experimentally determined scattering profile, and the size (4 m^2^) and albedo (0.3) of *Micius* given in Table 1, we are able to estimate the optical scattering cross section of a *Micius*-type satellite as
(11)$$\sigma_{sat}=4\times 0.3\times \sqrt{\cos\theta_{a}}.$$

For the scenario where excess photons scatter inside a satellite-borne QKD receiving optic and couple into the QKD detector, the scattering profile was found to be highly variable and dependent on the structure of the telescope. For example, the rim of the telescope optic was responsible for a significant fraction of detected photons. Scatter from objects placed near the receiving aperture greatly increased the excess photon rate. This means the effectiveness of this method of disruption is going to be highly dependent on the exact design of the telescope, and the design of the satellite. (For example, scatter of light off an antenna or other appendage on the satellite near the aperture will significantly increase number of photons coupled into the optical path of the receiver). This makes it difficult to make a good model for the internal scattering profile of the telescope.

For the 50 mm Galilean-type telescope used in these laboratory tests, a scatter profile of ${\left(\cos\theta_{a}\right)}^{2}$ gives a reasonable approximation for the scatter profile at angles of incidence greater than 5°. However, it will underestimate the number of detected photons at low angles of incidence, leading to a very conservative estimate of the disruption potential at these low angles (short distances of less than 100 km between the mobile laser terminal and the QKD ground station). A scaling factor of $10^{-7}$ is applied to the photon number, that is, the measurements show that only around one in $10^{7}$ photons incident on the QKD receiver optic from high off-axis angles couples into the SPD.

For a receiving optic of diameter $D_{s}$ with similar scattering characteristics to the Galilean telescope used here, a scattering profile of
(12)$$\sigma_{rec}=10^{-7}\times \frac{\pi D_{s}^{2}}{4}\times \cos^{2}\left(\theta_{a}\right),$$
gives a reasonable approximation for angles of incidence grater than 5°.

Because of the difficulty of developing a reliable scattering model that is applicable to a variety of different optics, detailed analysis of disruption of the satellite-side QKD receiver scenario will not be carried further in this paper. However, a general discussion will still be included because, since the laser terminal is directly targeting the receiving optic, rather than relying on scattered light returning to the ground, a significantly greater excess photon rate is likely to be achieved in this direct-targeting scenario.

### 3.2. Orbital Dynamics and Visibility

Regardless of the optical power level transmitted by the laser terminal, the exact location of the laser terminal relative to the ground station (though, more precisely, the location of the laser terminal relative to the closest point of the satellite’s ground track to the ground station) will determine the fraction of the QKD transmission period during which the laser terminal is able to view the satellite.

Figure 2 shows a contour plot where the contours indicate the fraction of the transmission window for which the laser terminal is able to see the satellite (at 500 km altitude) assuming a minimum elevation limit of 15°, similar to that of the *Micius* ground station. The ground track of the satellite is shown by the thin black line and is taken from an actual *Micius* ground track over Australia. The black dot represents the location of the hypothetical QKD ground station and the thick black bar shows the extent of the satellite’s path during which it is visible to the ground station and QKD can take place.

It can be seen that, depending on the exact location, the satellite is in view of a laser terminal 1000 km away for 50% to 60% of the QKD transmission window, and is in view for over 90% of the transmission duration to a laser terminal around 100 km from the ground station. If the elevation of the mobile laser terminal can be trained as low as 2°, then the laser terminal will be able to target the satellite through its entire transmission window. However, due to the large distance and atmospheric attenuation at these low angles, the excess photon rate will be significantly reduced.

It can also be seen that, depending on the exact positioning of the satellite ground track relative to the ground station, it is possible for the laser terminal to cover 100% of the QKD transmission window while still being located several tens of kilometres away from the ground station by positioning itself closer to the mid point of the QKD transmission window than the ground station is.

Figure 3 shows two example locations for a laser terminal. The left image shows the scenario for a laser terminal deployed 100 km to the east of the ground station, and the right image shows the terminal deployed 1000 km to the south east.

These orbital dynamics calculations were also used to determine the distance and angles between the ground station, satellite, and laser terminal, which were used to calculate the optical loss from laser terminal to QKD receiver.

### 3.3. Excess Photon Rate

Using the above results and following the calculations outlined in Section 2.4, we can estimate the excess photon rate at the QKD receiver for a given launch power from the laser terminal. We assume good seeing conditions and low atmospheric turbulence ($C_{n}^{2}=1\times 10^{-15}$ m^−2/3^) because these are the conditions under which *Micius* is currently capable of operating, and are the conditions under which many future QKD satellites will need to work and will achieve their highest quantum key rate.

The larger the transmitting aperture of the laser terminal, the more optical power will be incident on the satellite due to the reduced beam divergence. In this calculation, we have assumed that the laser launch aperture will have a diameter between 10 cm and 30 cm since these sizes are comparable to the optics currently used on a range of small-to-medium vehicle-mounted optical directed energy devices. (The calculations here further assume that the useable aperture of a telescope is roughly 80% of the diameter of the primary optic).

Figure 4 shows the estimated excess photon rates at the ground station for a laser terminal launching 1 kW of power with a 10 cm aperture 100 km away from the ground station, and Figure 5 shows the excess photon rate for the same launch power via a 30 cm aperture 1000 km away from the ground station. The laser power of 1 kW was chosen based on the optical loss from laser terminal, to satellite, to ground station, and the need to obtain certain excess photon rates at the ground station to achieve disruption. (The exact relative locations of the laser terminal and ground station in these example calculations correspond to the locations shown in Figure 3 above).

Atmospheric turbulence increases both the divergence of the laser beam and causes the direction of the beam to jitter. This significantly increases the optical losses. For example, for the scenario in Figure 5, increasing the ground-level turbulence to a moderate level of $C_{n}^{2}=1\times 10^{-14}$ m^−2/3^ reduces the excess photon rate at the ground station by about 60%. While this would significantly decrease the number of excess photons (or require a 3-fold increase in laser power to maintain the same received photon rate), the increased atmospheric turbulence will also significantly increase the losses on the QKD transmission, reducing the achieved quantum key rate, meaning fewer photons from the mobile laser terminal will be required to achieve complete disruption. (It should be noted that, over the 100 km and 1000 km distances involved in these calculations, it cannot be assumed that weather conditions are the same or comparable for ground station and laser terminal).

### 3.4. Quantum Bit Error Rate Due to Excess Photons

Applying the photon rates determined in Section 3.3 to the QBER calculation outlined in Section 2.5 we can estimate the effect the excess photons from the laser terminal will have on the QKD rate. Figure 6 shows additional QBER caused by the excess photons detected from a 1 kW laser terminal with a 10 cm telescope 100 km away, and Figure 7 shows the additional QBER due to a 1 kW laser terminal with a 30 cm aperture 1000 km away. In both figures, the green trace shows the detected QKD signal photon rate while the blue trace shows the detected excess noise photon rate. The dashed red line shows the QBER threshold at which no secret key can be produced and the QKD is completely disrupted.

Our analysis assumes that the laser terminal is emitting a continuous wave (CW) laser and that the excess photons from the terminal reaching the QKD receiver are uniformly temporally distributed. The red trace shows the additional QBER caused by the excess photons. Where this additional QBER exceeds the QBER threshold indicated by the dashed red line, no secure quantum key can be generated.

Figure 6 shows that, from 100 km away, the laser terminal can completely disrupt the QKD transmission for 70% of the duration of transmission window, and significantly increase the QBER over much of the remaining fraction. For the 1000 km case (Figure 7), the laser terminal is only able to completely disrupt the QKD for 55% of the transmission window, limited primarily by visibility of the satellite. A shown in Figure 8 second symmetrically placed laser terminal, such that the first terminal has sight of the QKD satellite in the early part of the transmission window, and the second terminal the later half would allow total disruption of the QKD transmission. Even with a reduction in received noise photons by a factor of two this symmetric attack would prohibit any secure key being generated during a satellite pass.

It should be noted that this analysis only considers the impact of the additionally created noise due to the attack. It does not consider the already existing noise due to other sources such as light pollution or sources of noise within the QKD system itself. Therefore, all disruption attempts will be more effective in a real-world scenario than estimated here as the QBERs due to other noise sources are cumulative.

## 4. Discussion

The results of this analysis show that substantial or complete disruption of modern satellite-mediated QKD can be achieved with simple equipment, that is, a kW-class laser, small telescope, and a mount able to track satellites. Continuous wave lasers or laser amplifiers capable of powers on the order of a few kW to tens of kW can be purchased commercial off-the-shelf for circa $100,000 USD. Optics on the order of 10 cm diameter that are able carry the resulting power densities are also commercially available. Telescope mounts able to accurately track satellites can be readily purchased for circa $20,000 USD. Combined, a malicious party aiming to disrupt a crucial satellite QKD link based on the technology demonstrated on board *Micius* would be able to do so with under $200,000 USD of equipment.

While the complete additional QBER calculation for the scenario where a satellite-borne QKD receiver is directly targeted by the laser terminal is not carried out here due to the lack of a reliable scattering model for this scenario, from the scattering experiments described in Section 2.2 which showed on the order of one in $10^{7}$ photons incident on the QKD optic couple into the QKD receiver, we expect that complete disruption of the QKD link in this scenario will be achieved with significantly lower laser powers than for the case where the laser power must scatter off the satellite and into the receiver at the ground station.

The ease with which such a laser terminal could be built and targeted at a satellite means that developers and users of future satellite QKD systems should take this threat into account. Based on our analysis, we discuss a number of countermeasures future QKD satellites can enact against this form of disruption.

### 4.1. Challenges Facing the Laser Terminal

The effectiveness of the laser terminal will be severely impacted by weather conditions, such as scattered or thin cloud. However, the QKD ground station is also reliant on good weather conditions. If cloud cover and other weather conditions near the ground station are suitable for effective QKD operation, then it is reasonable to assume that a laser terminal within 100 km of the ground station will be experiencing similarly suitable weather conditions. For the case where the laser terminal is around 1000 km away, cloud cover and other weather conditions could be very different. Deploying the terminal to a favourable position, or deploying multiple terminals (particularly in the case of scattered cloud) would be the only way to overcome adverse weather conditions. A malicious party would likely require multiple laser terminal to provide effective continuous, rather than sporadic, disruption from large distances from the ground station.

High ground will usually be more favourable for the laser terminal because it would allow the terminal to track the satellite at lower elevations, though the greatly increased loss through the atmosphere at low elevations will significantly reduce the excess photon rate.

Accurately locating the satellite will require accurate knowledge of the terminal’s own position. GPS will likely be more than adequate for accurate location of the laser terminal, however, satellites can be several hundred metres from their orbital track predicted from previous orbits. The laser terminal will have to acquire the satellite via scanning, optical tracking, or up-to-date information from radar or other tracking methods. A satellite such as *Micius* might be located relatively easily due to the transmission of tracking beacons, however, these could likely be phased out in future systems in order to counter the threat of disruption from simple laser terminals.

Pulsed lasers would greatly increase the effectiveness of the laser terminal by enabling greater disruption at lower average power levels (higher peak output for the same average power), but would require knowledge of the QKD detector gating. The extent to which it is possible to retrieve this information will need to be investigated, but could include monitoring the transmissions from the QKD transmitter.

### 4.2. Countermeasures

Our analysis has identified a number of countermeasures or design considerations a satellite QKD system could take to mitigate or reduce the impact of disruption due to ground-based laser terminals. Some of these design considerations are likely to be incorporated on improved satellite QKD systems in the future [8,17], and we discuss how they impact this form of disruption.

#### 4.2.1. Receiver Gating

Improved synchronization between the satellite and ground station will allow the QKD receiver to gate its detector more accurately, reducing the fraction of photons from the mobile laser terminal that will cause disruption. This would require the average power from the laser terminal to be increased, or better knowledge of the QKD receiver gating to be obtained.

This development is highly likely to occur in the near term as improved detector gating will be necessary to reduce the impact of background light and allow the QKD receiver to operate during the day or in light polluted areas. Great improvements in daytime operation of satellite laser ranging stations have already been achieved using careful gating techniques [18].

#### 4.2.2. Spectral Filtering

Another improvement likely to be incorporated into QKD systems is narrower spectral filtering of the incoming light, since this is also necessary to enable QKD receivers to operate in daylight. Narrower optical filters used to reject light not at the exact QKD transmission frequency will mean either better knowledge of the QKD transmission frequency will be required for effective disruption, or the laser terminal will have to emit light over a broader spectrum, increasing the average power consumption of the laser.

#### 4.2.3. Improved Sensitivity

While efforts to improve the sensitivity and efficiency of QKD systems will increase their effective secure bit rate, requiring a greater increase in QBER to achieve substantial disruption, many potential methods to increase the sensitivity of the QKD receivers to the signal from the QKD transmitter will also increase their sensitivity (and so, vulnerability) to photons from the laser terminal.

#### 4.2.4. Satellite Architecture

The QKD satellite could be designed in external shape and materials to minimize reflection and scattering of light towards the ground. This would required the laser terminal to increase its output power to achieve the same degree of disruption, though there may come a point where the laser power has to be increased to impractical levels.

The case where the QKD receiver is on board the satellite, satellite can be designed to shield its optics from off-axis exposure, minimize scattering surfaces such as antennae near the optic, and increase the internal baffling within the telescope.

#### 4.2.5. Higher Orbits

Future QKD satellites are likely to go to higher orbits (MEO or GEO) to increase the QKD transmission window, coverage area, and common-view range. This will also have the effect of increasing the propagation range and attenuation of the laser from the laser terminal. While the QKD system itself will also suffer additional attenuation, in the case where light scattered off the satellite is picked up by the QKD ground station, the light from the laser terminal suffers greater losses because it must make a round trip to the satellite and back to earth, unlike the one-way propagation of the QKD signal. Higher laser power will maintain the disruption, but the power of the laser from the mobile terminal will have to increase at a greater rate than the power of QKD transmission.

#### 4.2.6. QKD Protocols

Only the disruption of decoy state BB84 QKD has been analysed here. Other discrete-variable (DV-QKD) protocols and associated transmission architectures should be investigated including standard BB84, Ekert 91 [19], twin-field QKD [4], and likely future protocols.

Unlike DV-QKD, continuous variable QKD (CV-QKD) [20,21] modulates and measures the quadrature of the electric field of the light, rather than detecting the presence, or otherwise, of a single photon. This protocol is less mature than the most common DV-QKD protocols, but holds great promise because it can exploit conventional telecommunications hardware. Once developed sufficiently to deploy on board a satellite, CV-QKD will be able to apply much more stringent spectral filtering to the incoming light [22]. This will make it necessary to carefully target the correct transmission frequency, including adjusting for the Doppler shift due to the satellite’s motion. Uncertainty in the QKD transmission frequency will greatly increase the overall power required from the laser terminal in order to cover the expected frequency range with sufficient power to cause disruption.

## 5. Conclusions

We have shown that modern satellite-mediated QKD can be disrupted by noise caused by excess photons transmitted from a ground based laser terminal and scattered from the QKD satellite. The laser terminal needs a continuous launch power of around 1 kW to achieve complete disruption of a system based on that demonstrated by the *Micius* satellite. Lasers of this class are available commercially, and a malicious party could construct a simple laser terminal capable of causing significant disruption of QKD links at distances up to 1000 km for less than $200,000 USD. This means that such disruption is a significant security threat that should be taken into account in the design and use of future QKD satellite systems.

We developed a semi-empirical scattering model for the QKD satellite and performed orbital dynamics calculations to estimate the excess photon rate from a belligerent laser terminal reaching a QKD ground station up to 1000 km from the laser terminal. This photon rate is used to determine the additional QBER caused by the excess photons, which introduce noise into the QKD receiver.

We identify a number of countermeasures and design considerations future satellite QKD systems can implement to reduce the impact of this form of disruption. Likely future developments including moving to higher orbits and improved spectral and spatial filtering required to enable satellite QKD links to operate in daytime will improve the resilience of these links to disruption from remote lasers. However, all of these countermeasures simply require the laser terminal to increase its power on target to restore the same level of disruption.

We also discuss how this form of disruption could impact future satellite QKD technology. In particular, where the satellite includes an on board QKD receiver (the satellite supports a quantum memory or quantum repeater) will be even more vulnerable to disruption from a ground-based laser.

Other potential methods to disrupt satellite-mediated QKD link include mounting the laser terminal to an aircraft, scattering of noise photons from the atmosphere, and the generation and dispersal of scattering aerosols.

Our analysis shows these disruption methods pose a credible threat to satellite-mediated QKD in the future, and should be considered in the development and effective use of QKD satellite for high-value communications for areas such as finance, government operations, and defence. Additional analysis is required to determine the effects of this form of disruption on QKD protocols other than decoy-state BB84. The scattering and attenuation models used to predict the excess photon rate in this analysis could be verified by measuring the detected photon rate from bistatic satellite laser ranging experiments [23].

## Figures and Tables

**Figure 1 sensors-21-07904-f001:**
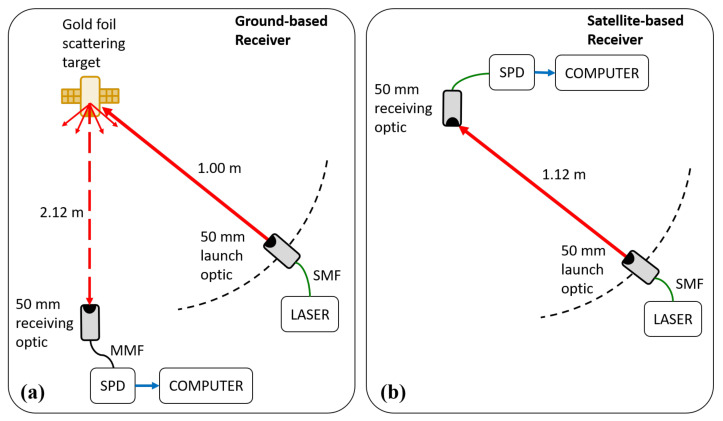
Simplified schematic of the setup to determine the optical scattering profile of a wrinkled gold foil target. Setup (**a**) simulates the scenario where light from the laser terminal scatters from the satellite and is detected by the QKD ground station, while setup (**b**) simulates the scenario where light from the laser terminal is detected by a satellite-based QKD receiver due to photons entering the receiving optic highly off-axis and scattering internally. SPD, single photon detector; SMF, single-mode fiber; MMF, multi-mode fiber.

**Figure 2 sensors-21-07904-f002:**
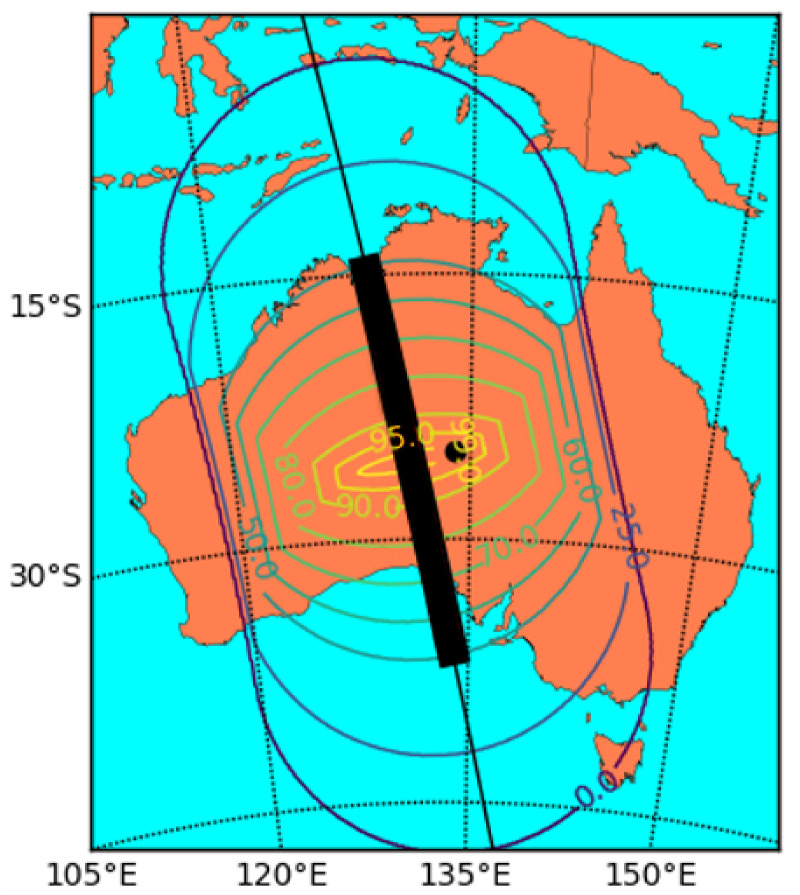
Contour plot showing the areas in which different fractions of the QKD transmission window are visible to the laser terminal. The black dot represents the QKD ground station, and the black bar represents the QKD transmission window.

**Figure 3 sensors-21-07904-f003:**
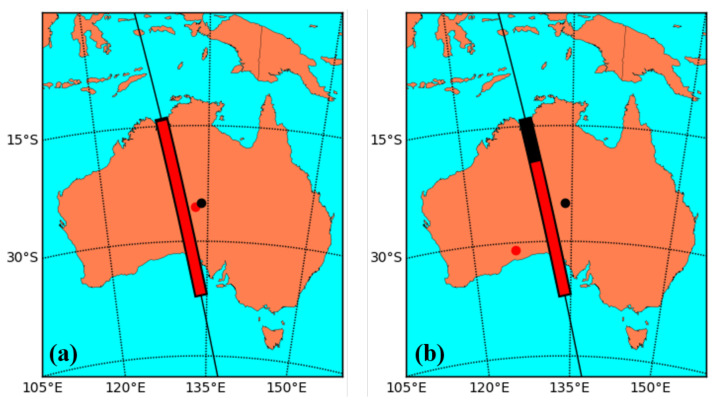
QKD transmission window coverage for laser terminals (**a**) 100 km away and (**b**) 1000 km away from the QKD ground station. The black dot represents the QKD ground station, and the black bar represents the QKD transmission window. The red dot represents an example location for the laser terminal, and the red bar shows the portion of the QKD transmission window that the satellite is visible to the laser terminal from this location.

**Figure 4 sensors-21-07904-f004:**
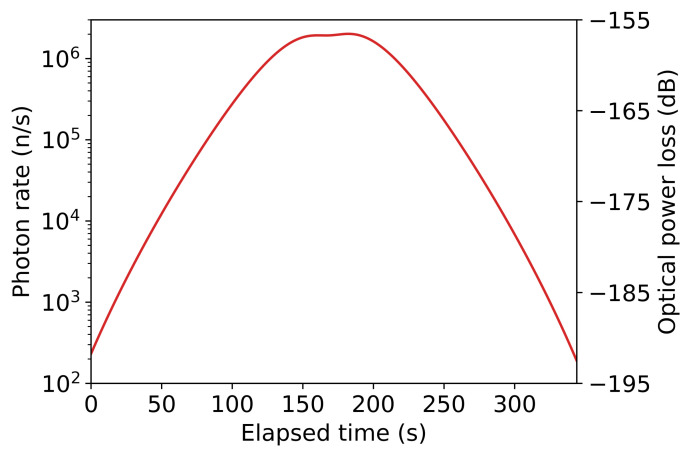
Excess photon rate at the QKD ground station and optical power loss over time (represented in seconds elapsed since the satellite came into view of the ground station) achieved for a 1 kW laser launched via a 10 cm telescope 100 km from the QKD ground.

**Figure 5 sensors-21-07904-f005:**
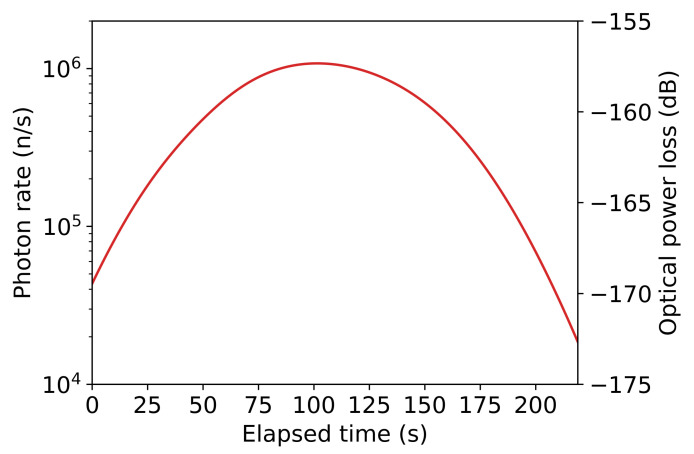
Excess photon rate at the QKD ground station and optical power loss over time (represented in seconds elapsed since the satellite came into view of the ground station) achieved for a 1 kW laser launched via a 30 cm telescope 1000 km from the ground station. (The satellite moves out of view of the laser terminal around 220 s assuming a lower elevation limit of 10°).

**Figure 6 sensors-21-07904-f006:**
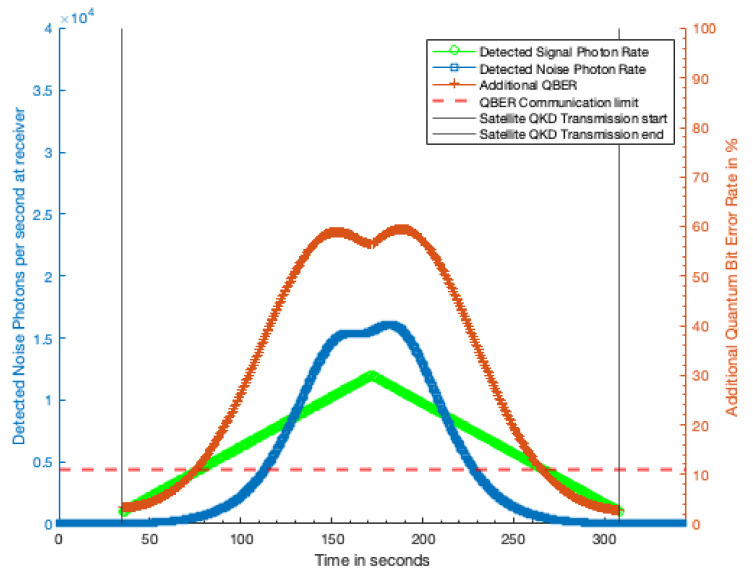
Additional QBER due to laser terminal with 10 cm optic from 100 km distance with 1 kW laser power. Green trace, detected QKD signal photon rate; blue trace, detected excess noise photon rate; red trace, additional QBER due to excess noise photons; dashed red line, QBER limit at which no secure key can be generated.

**Figure 7 sensors-21-07904-f007:**
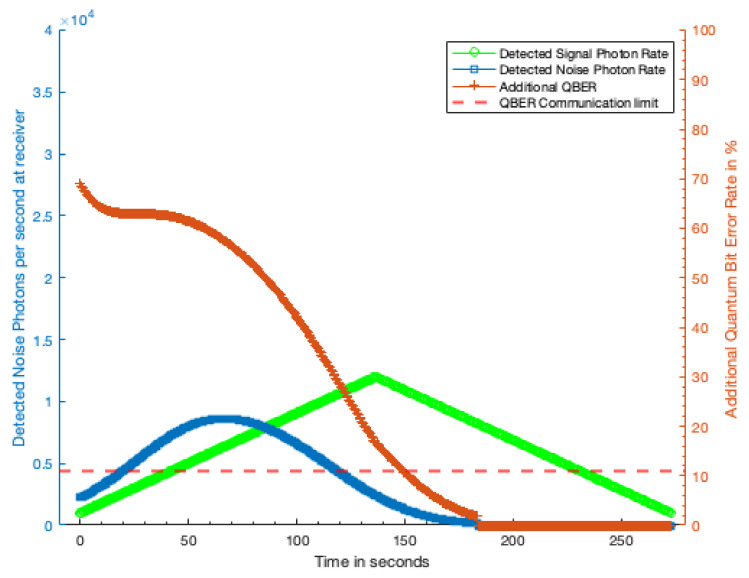
Additional QBER due to laser terminal with 10 cm optic from 1000 km distance with 1 kW laser power. Green trace, detected QKD signal photon rate; blue trace, detected excess noise photon rate; red trace, additional QBER due to excess noise photons; dashed red line, QBER limit at which no secure key can be generated.

**Figure 8 sensors-21-07904-f008:**
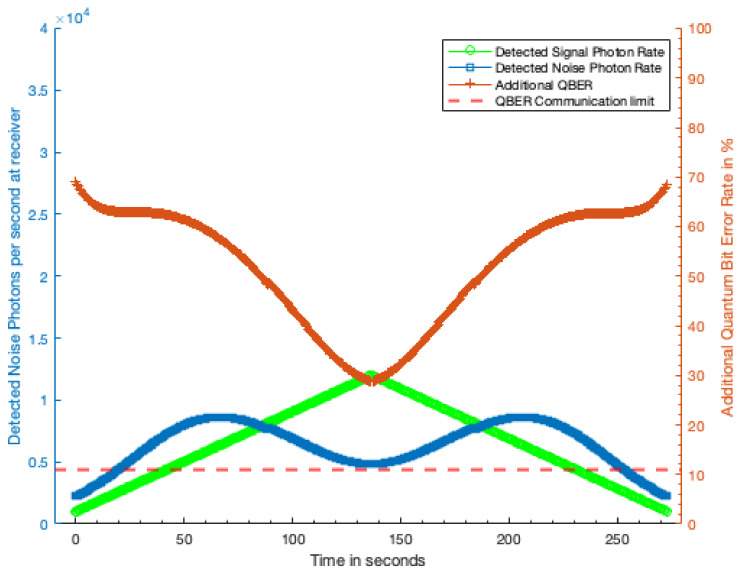
Additional QBER generated from two symmetrically placed 1 kW laser terminals both 1000 km from the receiving station so that one overlaps the early and the other the late part of the transmission window. At no point during the transmission window from the Satellite is any secure key generated as the additional QBER always exceeds 11%. Green trace, detected QKD signal photon rate; blue trace, detected excess noise photon rate; red trace, additional QBER due to excess noise photons; dashed red line, QBER limit at which no secure key can be generated.

**Table 1 sensors-21-07904-t001:** QKD satellite design assumptions based on *Micius*.

Parameter	Satellite Design
Orbit	Sun-synchronous circular orbit at 500 km altitude, 7.6 km/s. Satellite visible to ground station from elevations of approximately 15° to 10°.
Satellite bus	Approximately 2 m × 2 m, coated in metallized polymer thermal shielding with an albedo of approximately 0.3.
Satellite attitude	Coarse pointing — satellite is bodily oriented towards ground station with 0.5° precision.
Satellite QKD transmitter optic	300 mm aperture Cassegrain telescope. 10 μrad divergence, 22 dB diffraction loss at 1200 km.
Ground station QKD receiver optic	1 m aperture Ritchey-Chretien telescope. Approximately 16% optical efficiency from aperture to QKD receiver system.
QKD wavelength	848.6 nm
QKD protocol	Decoy-state BB84 [9] with three intensity levels.

## Data Availability

Data supporting this study can be obtained from the authors upon reasonable request.

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
