# Peer review of "Vulnerability of Satellite Quantum Key Distribution to Disruption from Ground-Based Lasers"

_sensors, 2021, doi:10.3390/s21237904_

Round 1

Reviewer 1 Report

The authors discuss the security of satellite-based quantum key distribution under active bright illumination attack in which the jamming light is sent by ground station, scattered by the satellite and received by the QKD receiver on the Earth. The number of jamming photons is estimated by analyzing the free space loss of ground-to-satellite-to-ground, scattering of satellite and orbital dynamics. Furthermore, quantum bit error rate can be obtained. Several countermeasures against this disruption are also suggested.

However, there are several comments the authors shall clarify.

(1) I prefer to suggest that the authors check Equation (9). Final key rate is usually calculated by subtracting the information eavesdropped by attackers from the mutual information between authenticated two participants. In the first paragraph of Section 2.5, “this threshold is at 0.11, or 11%”. It seems that the result can not be obtained according to Eq. (9).

(2) The left side of Eq. (4) might be theta_j.

(3)  What is the value of theta_p in Eq. (1).

Author Response

The authors would like to thank the reviewer for their helpful suggestions and feedback. We have implemented changes and clarifications based on each point.

(1) I prefer to suggest that the authors check Equation (9). Final key rate is usually calculated by subtracting the information eavesdropped by attackers from the mutual information between authenticated two participants. In the first paragraph of Section 2.5, “this threshold is at 0.11, or 11%”. It seems that the result can not be obtained according to Eq. (9).
The reviewer is correct in that Eq 9 does not lead to the absolute threshold of 11%. Instead Eq 9 is what is used to calculate the net key rate from a QKD system that can be obtained in light of all the considered imperfections in the privacy amplification and distillation steps. If all others systems are perfect and a pure single photon source is used the maximal threshold was derived by Shor and Preskill (new citation 12). We have added this reference to the manuscript.
The first paragraph of section 2.5 has been modified to clarify this: For the BB84 protocol used by Micius, this threshold is at 0.11, or 11% [12] and is further reduced if imperfect sources and detections systems [13,14] are considered as all leaked information has to be considered. For our investigation we will focus on the fundamental limitation of an 11% QBER which is independent of any specifics of the utilised equipment or operating conditions.

(2) The left side of Eq. (4) might be theta_j.
The reviewer is correct. The equation has been modified to show theta_j.

(3) What is the value of theta_p in Eq. (1).
For this analysis we assume zero pointing error. This is reasonable because the optical losses will be dominated by the divergence and pointing jitter, and this is a feasibility study, so neglecting minor pointing errors will not significantly affect the final laser power and disruption level estimates.
Paragraph 6 in section 2.3 has been modified to clarify this point: For the purposes of this analysis we assume that the laser terminal has very good tracking and the loss due to pointing error, theta_p, is much smaller than those due to divergence and pointing jitter, so we set theta_p=0.

Reviewer 2 Report

The paper shows that a ground-based laser with the power of ~1 kW can significantly disrupt modern satellite QKD systems due to photons scattering off the satellite being detected by the QKD receiver on the ground. Overall,this work is interesting and helpful to the practical application of satellite-based qkd in the future. However, it is unclear whether their claim is properly justified. There are some points that should be further considered.

  1. In section 3.2, Figure 1 and Figure 2(b) are the same. Maybe the two figures should be merged.
  2. In section 2.2, the authors describe the method and experimental setup for scattering measurements, but there are no experimental schematics and details, which should be added. And how this simulation experiment can compare with the satellite scattering situation also needs a clarification.
  3. In section 2.5, the authors say, “For the BB84 protocol used by Micius, this threshold is at 0.11, or 11%.” In fact, the ideal single photon in the experiment is very difficult to achieve, and the bb84 protocol with the decoy state used by Micius. The threshold is about 4-5% and this change directly affects the laser power. The authors should revise the value of the threshold of QBER.
  4. In section 3,the author directly provide the result of the 1kw. I did not see the available describing how this 1kw is calculated. I think this point is not very clear.
  5. Throughout the paper,only power parameters are given for the ground-based laser, without mentioning whether the laser spatial mode is single-mode or multi-mode, and using a laser with a multi-mode in ground-to-satellite transmission significantly increases energy attenuation. In addition, I think the safety issues of high-power laser also need to be considered.
  6. The ground-based laser tracking satellites require orbit prediction,and the paper does not mention the requirement of the precision of the actual orbit prediction. The precision of orbit prediction will affect the design of laser divergence angl Too large the divergence angle introduces additional energy attenuation, and too small is likely to fail to track the satellite.
  7. In conclusion, the authors say, “We also discuss how this form of disruption could impact future satellite QKD technology. In particular, where the satellite includes an on board QKD receiver (the satellite supports a quantum memory or quantum repeater) will be even more vulnerable to disruption from a ground-based laser.” I think this statement is not very reasonable. The receiving field of the satellite is very small. The disrupting laser beam launched from the laser terminal from the ground station will be very difficult to enter the detector on the satellite. Therefore, I think this disruption has little effect on the QKD receiver

Author Response

The authors would like to thank the reviewer for their thorough and helpful feedback which, we feel, has helped to improve the clarity of the manuscript.

1.  In section 3.2, Figure 1 and Figure 2(b) are the same. Maybe the two figures should be merged.
Figure 1 (now figure 2 due to the addition of a new figure, see point 2) has been modified. The red dot indicating the ground station and the red bar indicating the disruption window have been removed from this figure and are now only shown in the later figure. (Former) Figure 1 now only shows the disruption visibility contours and not the disruption window, while the following figure is used to show specific examples at 100 km and 1000 km distance.

2. In section 2.2, the authors describe the method and experimental setup for scattering measurements, but there are no experimental schematics and details, which should be added. And how this simulation experiment can compare with the satellite scattering situation also needs a clarification.
A simplified schematic of the experimental setups has now been added to section 2.2. The first two paragraphs of section 2.2 explain how this experiment is comparable to the satellite scattering scenarios, and we believe the addition of the experimental schematic has made this significantly clearer.

3. In section 2.5, the authors say, “For the BB84 protocol used by Micius, this threshold is at 0.11, or 11%.” In fact, the ideal single photon in the experiment is very difficult to achieve, and the bb84 protocol with the decoy state used by Micius. The threshold is about 4-5% and this change directly affects the laser power. The authors should revise the value of the threshold of QBER.
The reviewer is correct that the actual threshold for the protocol used with Micius in reference 7 is much lower which is evident in their minuscule keyrate in comparison to the number of received photons. However, as this manuscript looks at the conceptual method of disrupting BB84 style QKD with satellites we consider it more prudent to compare to an ideal transmission system and look for the requirements of disruption in such a case. Real world disruption will be attained most likely with lesser requirements of the disrupting equipment.
The first paragraph of section 2.5 has been modified to clarify this: For the BB84 protocol used by Micius, this threshold is at 0.11, or 11% [12] and is further reduced if imperfect sources and detections systems [13,14] are considered as all leaked information has to be considered. For our investigation we will focus on the fundamental limitation of an 11% QBER which is independent of any specifics of the utilised equipment or operating conditions.

4. In section 3, the author directly provide the result of the 1kw. I did not see the available describing how this 1kw is calculated. I think this point is not very clear.
Paragraph 3 in section 3.3 has been modified to clarify this: The laser power of 1 kW was chosen based on the optical loss from laser terminal, to satellite, to ground station, and the need to obtain certain excess photon rates at the ground station to achieve disruption.

5. Throughout the paper, only power parameters are given for the ground-based laser, without mentioning whether the laser spatial mode is single-mode or multi-mode, and using a laser with a multi-mode in ground-to-satellite transmission significantly increases energy attenuation. In addition, I think the safety issues of high-power laser also need to be considered.
Paragraph 4 in section 2.4 has been modified to clarify the assumptions about laser mode and quality: In this analysis, we will assume the laser terminal launches a single-mode Gaussian beam. This is a reasonable assumption because modern commercial off-the-shelf lasers are able to launch on the order of several kilowatts of power with beam quality factors near unity.
The aim of this paper is to perform a feasibility study on disruption of satellite QKD, and we have found that this method of disruption is feasible for malicious parties with a few hundred thousand USD in resources. Because of this, safety issues related to the use of high-power lasers are outside the scope of considerations for this work.

6. The ground-based laser tracking satellites require orbit prediction, and the paper does not mention the requirement of the precision of the actual orbit prediction. The precision of orbit prediction will affect the design of laser divergence angle. Too large the divergence angle introduces additional energy attenuation, and too small is likely to fail to track the satellite.
Because this is a feasibility study, and there are a huge range of engineering options available to the laser terminal to achieve good tracking, we have elected to analyse the scenario where pointing error is much smaller than pointing jitter and beam divergence. This is now noted in paragraph 6 in section 2.3: For the purposes of this analysis, we assume that the laser terminal has very good tracking and the loss due to pointing error, theta_p, is much smaller than those due to divergence and pointing jitter, so we set theta_p=0.
We believe further discussion of this point is outside the scope of the paper. However, we note, in section 4.1, that there is a range of options for achieving sufficiently high-precision tracking of the satellite, including optical tracking using the satellite’s transmissions or when the satellite is illuminated by the sun, real-time information fed in from radar, or a variety of techniques used in daytime laser ranging of satellites and space debris, including those discussed in reference [18].

7. In conclusion, the authors say, “We also discuss how this form of disruption could impact future satellite QKD technology. In particular, where the satellite includes an on board QKD receiver (the satellite supports a quantum memory or quantum repeater) will be even more vulnerable to disruption from a ground-based laser.” I think this statement is not very reasonable. The receiving field of the satellite is very small. The disrupting laser beam launched from the laser terminal from the ground station will be very difficult to enter the detector on the satellite. Therefore, I think this disruption has little effect on the QKD receiver
We find that on the order of one in 10^7 photons incident on the face of the receiver optic will couple into the QKD receiver in this scenario, meaning total disruption is much easier to achieve than for the ground-based receiver, and this is an important scenario to consider.
As we discuss in section 2.2, section 2.4 (eqtn 2), and section 3.1 (eqtn 12), our analysis also considers the scenario where photons from the laser terminal couple into a satellite-borne receiver due to photons scattering within the receiver optics having entered from a high off-axis angle. While (as noted in section 3.1) we are not able to make consistent predictions for this scenario due to the large degree of variability of scattering within the optics, this will have little dependence on the receiving field of the optic, and means there is a much greater flux of photons adding noise to the satellite-borne QKD receiver than for the ground-based receiver. As we note in section 4.2.4, attention to baffling within the receiver optics is important to help mitigate disruption in this scenario.

Reviewer 3 Report

This paper can be accepted

Author Response

The reviewer did not identify make any suggestions for change or identify any issues with the manuscript. The authors would like to thank the reviewer for their consideration of our paper.

Round 2

Reviewer 2 Report

The authors have addressed all the issues my concerned.